# UniArt: Generating 3D articulated objects with open-set articulation beyond retrieval

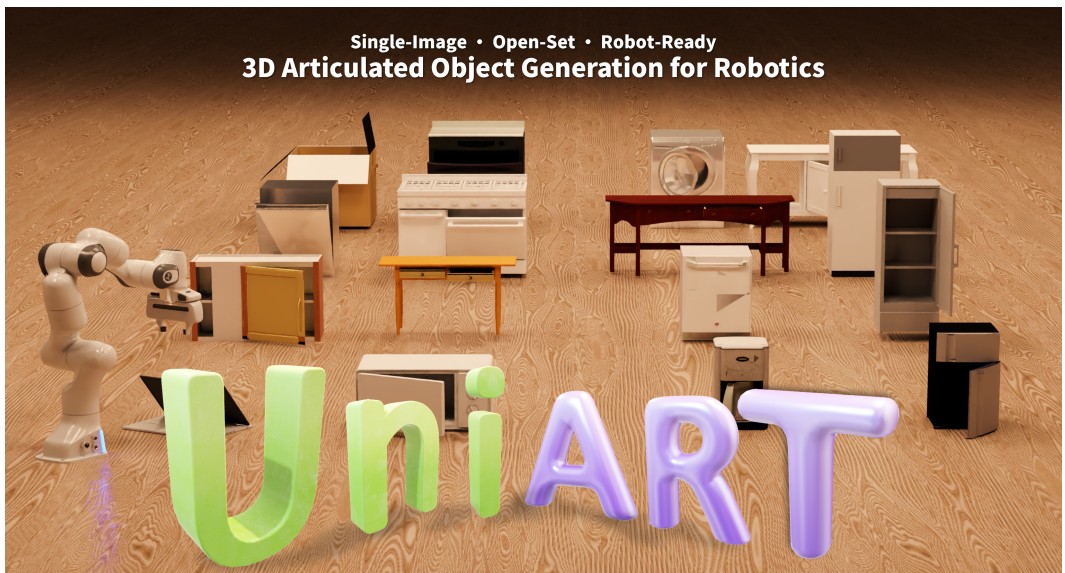

Figure 1: We propose UniArt, the first non-retrieval, diffusion-based framework that generates robot-ready articulated 3D objects from a single image, enabling open-set generalization for scalable simulation and manipulation.

## Abstract

Articulated objects are central in the field of realistic simulation and robot learning, enabling dynamic interactions and task-oriented manipulation. However, manually annotating these objects is labor-intensive, motivating the need for automated generation solutions. Previous methods usually rely on retrieving part structures from existing datasets, which inherently restricts diversity and causes geometric misalignment. To tackle these challenges, we present UniArt, an end-to-end framework that directly synthesizes 3D meshes and articulation parameters in a unified manner. We decompose the problem into three correlated tasks: geometry generation, part segmentation, and articulation prediction, and then integrate them into a single diffusion-based architecture. By formulating both part segmentation and joint parameter inference as open-set problems, our approach incorporates open-world knowledge to generalize beyond training categories. We further enhance training with a large-scale, enriched dataset built from PartNet-Mobility, featuring expanded part and material diversity. Extensive evaluations show that UniArt substantially outperforms existing retrieval-based methods in mesh quality and articulation accuracy, especially under open-set conditions. Code will be publicly available to foster future research in the 3D generation and robotics societies.

## 1 Introduction

3D articulated objects Quigley et al. (2015) are core components of mechanical systems, ranging from common doors in daily life to complex joint mechanisms in robotic grippers. Unlike static rigid

bodies, articulated objects exhibit inherent part-level structures and motion patterns, enabling dynamic interactions such as opening a drawer, swiveling a chair, or operating scissors. Precise modeling of these structures (Tseng et al., 2022; Liu et al., 2023a; Weng et al., 2024; Iliash et al., 2024; Mandi et al., 2024; Liu et al., 2024a;b; Wu et al., 2025a) not only supports the development of high-fidelity simulation environments (Chen et al., 2024a; Li, 2023; Li et al., 2024b; Luo et al., 2023) but also paves the way for accurate dynamic analysis in embodied robotics (Yang et al., 2024a;b; Geng et al., 2025). However, acquiring detailed annotations for such objects remains highly labor-intensive and struggles to keep pace with growing object diversity, thus driving the need for automated generation solutions.

Existing methods for articulated object generation, such as those described in (Liu et al., 2024a;b; Wu et al., 2025a; Deng et al., 2024), generally follow a three-stage pipeline. First, articulation parameters, including part bounding boxes and semantic labels, are predicted from input images. Next, a corresponding part geometry is retrieved from a pre-existing asset library. Finally, the retrieved parts are assembled into a complete object. While retrieval-based methods Jiang et al. (2022); Gao et al. (2025); Su et al. (2025); Qiu et al. (2025); Liu et al. (2025b); Shen et al. (2025) provide a shortcut for generating new articulated models, they introduce several critical limitations: geometric misalignment due to imperfect part matching, limited diversity bounded by the pre-defined asset collection, and poor generalization to objects outside the training distributions. These issues hinder the deployment of such methods in open-world scenarios, where objects exhibit vast variations in form, function, and material.

To address these crucial issues, we propose UniArt, an end-to-end framework that synthesizes articulated objects directly without relying on part retrieval. Our paradigm shift centers on rethinking two core concepts. **First**, we reformulate the task as a conditional generation of 3D assets with multi-faceted features encompassing geometry, appearance, part segmentation, and articulation structure. Specifically, UniArt encodes these attributes into a unified latent representation, named UniArt latents, and jointly generates both shape and motion parameters within a unified diffusion-based architecture. **Second**, we treat articulation type prediction as an open-set problem, eliminating the need for predefined joint semantic labels during training. This approach significantly enhances generalization beyond the training categories.

To support effective learning, we also compile a large-scale dataset based on PartNet-Mobility, augmented with diverse part geometries and material properties. Comprehensive evaluations on the PartNet-Mobility benchmark demonstrate that UniArt outperforms existing baselines significantly in terms of mesh quality and articulation accuracy, particularly under challenging open-set conditions.

Our contributions can be summarized as follows:

- We reformulate the articulated object creation task as a conditional generation task, where the input is a single image and the output is an articulated object with high-fidelity geometry, well shape-image consistency, and precise articulation.
- We introduce UniArt latent representations that jointly encode object geometry, appearance, part segmentation, and articulation parameters within a diffusion-based architecture..
- We formulate articulation prediction as an open-set problem, removing dependency on fixed joint semantics and significantly improving generalization to unseen object categories.
- We show through comprehensive experiments that our method substantially advances the state of the art in articulated object generation.

## 2 RELATD WORKS

### 2.1 RECONSTRUCTION-BASED ARTICULATED OBJECT CREATION.

The reconstruction methods (Tseng et al., 2022; Liu et al., 2023a; Weng et al., 2024; Iliash et al., 2024; Mandi et al., 2024; Kim et al., 2025) typically rely on multi-view or multi-state inputs to recover part-level geometry and articulation parameters. On the basis of NeRF, CLA-NeRF (Tseng et al., 2022) utilizes a component segmentation field to predict the categories of each component of the articulated object, in order to perform view synthesis, component segmentation, and joint pose estimation of unknown articulated poses. PARIS (Liu et al., 2023a) presents a self-supervised

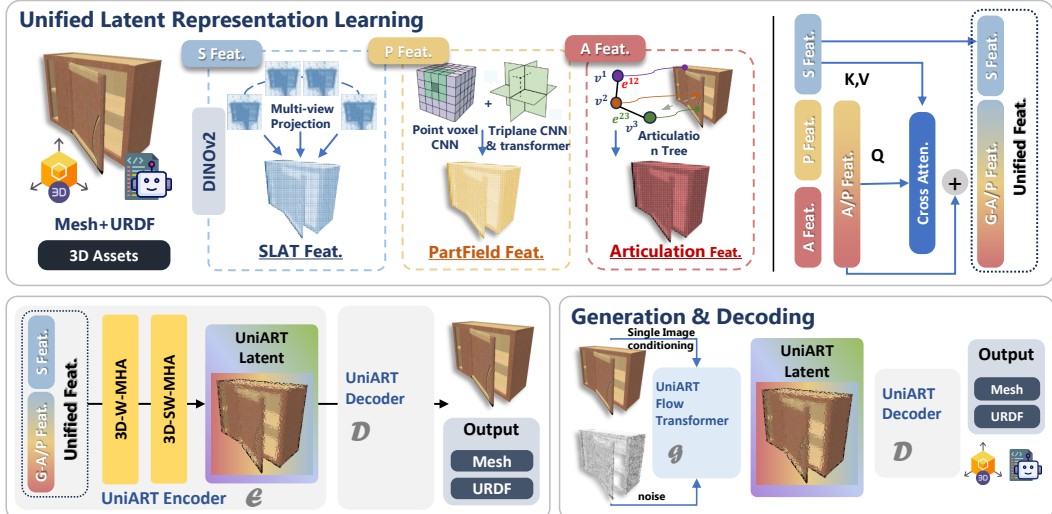

Figure 2: Overview of **UniArt**. Unified latent representation learning combines geometry, parts, and articulation features. A diffusion-based generator then decodes them into robot-ready articulated 3D meshes and URDFs from a single image.

architecture for part-level reconstruction and motion analysis of articulated objects, achieving significant improvements in shape reconstruction and motion estimation without requiring 3D supervision. Real2Code (Mandi et al., 2024) utilizes the knowledge of LLM to get the articulation parameters, also requiring multi-view images as input. ArtGS (Liu et al., 2025b) employs a strategy from coarse to fine, using the Hungarian algorithm to match Gaussian spheres in different states and cleverly establish corresponding relationships between different states of objects. GaussianArt (Shen et al., 2025) instead introduces a unified representation based on articulated 3D Gaussian primitives, generating good reconstruction results with correct articulation parameters. While reconstruction-based methods provide good results for generating new articulated objects, they rely on multi-view or multi-state inputs, which are not easily accessible for scalable data construction. In contrast, our method requires only a single image as input. This substantially reduces input complexity, enabling scalable data collection and facilitating large-scale robot training.

## 2.2 GENERATIVE 3D ARTICULATED OBJECT CREATION.

Recent progress in 3D generation (Li et al., 2025; Chen et al., 2025; Li et al., 2024a; Ren et al., 2024; Tochilkin et al., 2024; Wang et al., 2023; Zhao et al., 2025; Wu et al., 2024; 2025b) has enabled applications in 3D Articulated Object Creation. The generative 3D articulated object creation aims to generate part-level geometry and articulation parameters through a single image. Previous generative articulated object creation methods typically rely on retrieval, where a corresponding part geometry is retrieved from a pre-existing asset library. Articulate-Anything (Le et al., 2024) first converts static 3D assets into articulation-ready models and has sparked growing interest in generating objects equipped with URDF-style hinge joints. It retrieves the most similar asset from the library through CLIP similarity and generates articulation parameters through reinforcement learning. URDFormer Chen et al. (2024b) attempts to directly infer an interactive URDF from the images. Likewise, OPDMulti (Sun et al., 2024) localizes movable parts and estimates motion parameters from a single image. Le et al. (2024); Liu et al. (2024a;b); Wu et al. (2025a) generate articulated object structures from inputs such as images and graphs. However, these methods often depend on mesh retrieval from a fixed database, which restricts both the variety of generated objects and the adaptability to subjective user specifications. Compared with these, our generative approach aims to synthesize new objects directly rather than recover articulation from existing geometry.

## 3 PROBLEM FORMULATION

Existing retrieval-based pipelines approach articulated object generation by first predicting part proposals, then searching a finite repository of pre-rigged assets for the closest matches, and finally

stitching the retrieved parts together. Due to the fact that each component is copied verbatim from the database, these methods cannot guarantee geometric continuity at part boundaries, inherit whatever material and joint types the database happens to offer, and fail gracefully when the target object falls outside the pre-defined taxonomy. In contrast, we reformulate the task as a fully generative problem, synthesizing geometry, part structure, and articulation parameters in a continuous latent space so that every piece is produced with mutual consistency and the design space is no longer bounded by the content of a repository.

Formally, we denote the target articulated asset as:

$$A = (M, S, U) \tag{1}$$

where $M$ is a watertight triangle mesh, $S$ is a part segment mask, and $U$ is a URDF specification (Quigley et al., 2015) containing joint type $T$, connection topology $J$, axis direction $A$, joint limits $L$ and body assignments $B$ of each part:

$$U = (T, J, A, L, B) \tag{2}$$

The input is a single RGB image $I$.

Retrieval-based methods first infer an articulation tree from the given image $I$, where each part contains the bounding boxes $B = \{b, c\}$ that list each part's 3D bounding box $b \in \mathbb{R}^6$ and semantic label $c$. They then select parts from a part repository $\mathcal{P} = \{(M_i, \ell_i)\}_{i=1}^N$ that provides mesh geometry $M_i$, and the corresponding label $\ell_i$, the method filters the database by label, $\mathcal{D}_c = \{i \mid \ell_i = c\}$, and retrieves the closest candidate by box similarity,

$$\hat{M} = \arg \min_{i \in \mathcal{D}_c} d_{\text{size}}\big(\text{box}(M_i), b\big). \tag{3}$$

The selected parts are rigidly aligned to the predicted boxes and concatenated to yield

$$A^{\text{retr}} = \mathcal{A}\big(\{\hat{M}_k\}_{k=1}^K, B\big) = (M^{\text{retr}}, S^{\text{retr}}, U^{\text{retr}}). \tag{4}$$

where $K$ is the number of parts in the articulation tree. It is observed that every component of $A^{\text{retr}}$ is copied from $\mathcal{D}$, the output can never exceed the geometric fidelity, material diversity, or articulation vocabulary encoded in the repository, and inevitable misalignments across part boundaries produce visual and kinematic inconsistencies.

Instead, in the generative method, we learn a conditional diffusion model over a single latent vector $z \in \mathbb{R}^d$, dubbed UniLatent, that jointly encodes geometry, appearance, part structure, and the articulation tree. The forward process corrupts $z$ with Gaussian noise, while the reverse process produces $z_0 \sim p_\theta(z \mid I)$. A shared decoder then deterministically maps $z_0$ into an articulated asset $A = (M, S, U)$ through three parallel heads $G_{\text{geo}}, G_{\text{seg}}, G_{\text{art}}$, formulated as:

$$(M, S, U) = \big(G_{\text{geo}}(z_0),\ G_{\text{seg}}(z_0),\ G_{\text{art}}(z_0)\big), \tag{5}$$

yielding the overall distribution

$$p_\theta(A \mid I) = \int \delta\big(A - G(z)\big)\, p_\theta(z \mid I)\, dz, \tag{6}$$

where $G = (G_{\text{geo}}, G_{\text{seg}}, G_{\text{art}})$ and $\delta(\cdot)$ is the Dirac delta. In this setting, $p_\theta$ is learned in a continuous latent space where UniArt can produce infinitely many geometries and articulation patterns that are not restricted to the discrete set $\mathcal{D}$, while its open-set formulation removes the need for fixed semantic labels during training and enables robust performance on previously unseen categories.

## 4 METHODS

As illustrated in Fig. 2, our goal is to generate articulated objects in a unified framework that simultaneously produces geometry meshes, part-level segmentation, and articulation parameters. To support this generation process, we should parameterize the geometry, part segment and articulated structure into vectors that can be the target of the diffusion. We introduce how we parameterize the articulated objects into a latent space in Sec . 4.1. Then we introduce our variational autoencoder that compresses the parameterized articulated object into the latent space in Sec . 4.2. Finally, we illustrate the generation process in Sec . 4.3.

### 4.1 Articulated Object Parameterization

As mentioned before, we adopt the URDF representation for parameterization of articulated objects, which represents each object as a connected graph in which nodes denote links (parts) and edges denote joints. We follow the common practice of NAP (Lei et al., 2023) and assume the kinematic graph is connected with no cycles and each edge is a screw joint with at most one prismatic translation and one revolute rotation, covering most real-world articulated objects (Xiang et al., 2020; Wang et al., 2019).

A central challenge in parameterizing articulated objects is how to encode the joint-level kinematics in a representation that is spatially compatible with voxelized geometry. While URDF specifies each part node and its connecting joints in symbolic form, the geometry mesh is voxelized into a dense feature grid. These attributes must be grounded into a continuous volumetric tensor for unified encoding. We address this via a joint-to-voxel embedding scheme.

We describe the URDF parameters $U = \{u^0, u^1, ..., u^{K-1}\}$ as a graph with $K$ nodes and each $u_i$ attributes encoding joint type $t^i$, axis $a^i \in \mathbb{R}^6$ and motion limits $l^i = (l_{\min}, l_{\max})$, denoted as: $u^i = (t^i, a^i, l^i)$. Unlike traditional retrieval-based approaches that often rely on predefined semantic labels of links (like base, door, drawer, handle, knob, tray in SINGAPO (Liu et al., 2024a)), we intentionally exclude such categorical annotations in our formulation. This design choice avoids introducing bias toward a fixed set of link categories and instead encourages the model to generate links and kinematic structures that are not limited to predetermined templates, thereby improving generalization to novel articulation morphologies. We serialize $U$ into a sparse adjacency tensor $c \in \mathbb{R}^9$, which serves as the articulation representation of a joint. For the connection graph $J$, we form the adjacency tensor $J \in \{0, 1\}^{K \times K}$, which serves as an attention mask to guide the articulation generation.

Then we conduct joint-to-voxel projection, which aligns the joint parameters with the corresponding mesh and part structure. For each node $i \in \{0, \ldots, K-1\}$, we associate its attributes to the edge that connects this node to its parent in the kinematic tree. This design ensures that the parameters are naturally interpreted as governing the motion of the child link with respect to its parent link.

The attributes carried by the edge are then projected onto the 3D voxel space that represents the geometric occupancy of the child part. In this process, all active voxels belonging to the mesh region of the child part inherit the same parameter assignment, thereby embedding the kinematic constraints directly into the spatial representation of the part. This provides a unified voxel-level representation where both geometric and kinematic information co-exist, enabling subsequent models to jointly reason about structure and motion. Since each node in the articulation tree has exactly one parent, this assignment is reversible during decoding. Given voxel-level encodings, we can uniquely recover the corresponding node attributes and rebuild the parent-child relationships. This property is essential to guarantee consistency between the learned voxelized representation and the original kinematic graph structure.

### 4.2 UniArt VAE with Geometry-Articulation Interaction

After obtaining the voxelized representation of both geometry and articulation, our next step is to learn a compact latent space that jointly captures structural and kinematic information. We construct a unified structured latent representation, named UniArt Latent, and utilize a variational autoencoder (VAE) tailored for this unified representation.

For each 3D asset, we first convert the mesh into a binary occupancy grid, resulting in a voxelized geometric feature $V_{geo}$ enriched with visual features by multiview average, following Xiang et al. (2025). In parallel, the articulation representation introduced in the previous subsection is also voxelized, producing per-voxel articulation attributes. For the part representation, we utilize a pretrained model partfield Liu et al. (2025a) to generate part-aware representations. The part representations and articulation attributes are added and voxelized, resulting in final articulation features $V_{art}$. All features are defined on the same voxel space with size $N$, where $N$ represents the total number of active voxels.

Instead of relying on naive concatenation, we introduce an attention block to dynamically align $V_{geo}$ and $V_{art}$. Specifically, we treat the articulation feature as the query and the geometric feature as

key–value pairs:

$$F_{art} = \text{Attention}(Q = V_{art}, K = V_{geo}, V = V_{geo}) + V_{art}, \tag{7}$$

where the cross-attention modules aggregate motion-aware features that are consistent with the underlying geometric structure. The fused representation is enhanced with a residual connection to preserve original geometric detail. These two feature types are channel-wise concatenated into a unified voxel feature map:

$$V = \text{Concat}(V_{geo}, F_{art}) \tag{8}$$

where each voxel is enriched with both spatial occupancy and articulation-aware information. This design ensures that articulation information is selectively integrated depending on local geometry, encouraging the model to learn physically plausible correlations between part shape and its kinematic behavior.

The unified feature $V \in \mathbb{R}^{N \times C}$, with $N$ active voxels and $C$ channels, is then passed through the VAE encoder $\mathcal{E}_{vae}$. The encoder employs attention layers to learn hierarchical spatial features while preserving the alignment between geometry and motion constraints. The sampled latent embedding $z$ is passed into the VAE decoder $\mathcal{D}_{vae}$, which reconstructs both geometry and articulation features simultaneously:

$$\hat{V}_{geo}, \hat{V}_{art} = \mathcal{D}_{vae}(z). \tag{9}$$

Unlike conventional VAE frameworks (Cao et al., 2025) that separately encode physical or appearance properties, our decoder is optimized to jointly restore the voxelized structure and articulation. This design ensures that the model learns a latent space where geometry and motion are inherently entangled, facilitating more faithful morphology reconstruction.

The VAE is optimized with a compound loss function:

$$\mathcal{L}_{vae} = \mathcal{L}_{geo} + \mathcal{L}_{art} + \mathcal{L}_{kl}. \tag{10}$$

where $\mathcal{L}_{geo}$ measures the reconstruction fidelity of voxel occupancy, $\mathcal{L}_{art}$ supervises the recovery of articulation attributes, and $\mathcal{L}_{kl}$ is the Kullback–Leibler regularization term. Together, these terms encourage the VAE to disentangle structural and kinematic variations while maintaining a compact latent space suitable for downstream generation and inference tasks.

### 4.3 ARTICULATED LATENT GENERATION

After obtaining the fused latent representation from the VAE encoder, we aim to generate novel articulated objects with consistent geometry and articulation. We design a latent diffusion model that simultaneously models structural layout and articulation parameters. The generator is implemented as a rectified flow mod- el, similar to Xiang et al. (2025), and the training objective is the conditional flow matching objective: $\mathcal{L} = \mathbb{E}_{t,x_0,\epsilon} ||f(x,t) - (\epsilon - x_0)||_2^2$ where $f(x,t)$ is the conditional flow field that transports noisy samples to the clean latent distribution, $x_0$ is a latent from the VAE encoder, $\epsilon$ is Gaussian noise, and $t$ is the timestep.

## 5 EXPERIMENTS

We evaluate UniArt on the PartNet-Mobility benchmark, which provides a diverse set of articulated objects with ground-truth meshes, part annotations, and URDF parameters. Besides the common evaluation practice, we also conduct open-set evaluation. We split the dataset into seen categories (Storage, Table, Refrigerator, Dishwasher, Oven, Washer, and Microwave) and unseen categories (Bottles, Toilet, Chair, etc.) to test open-set generalization.

### 5.1 EXPERIMICAL SETUP

We follow the dataset split utilized in common evaluation practice (Wu et al., 2025a; Liu et al., 2024a). During training, we augment datasets with random perturbations in part geometry, synthetic articulation parameter sampling within physically valid ranges. The total training samples are 45k. We utilize the AdamW optimizer with a learning rate of $1e-4$. Models are trained on 8 NVIDIA A100 GPUs with a batch size of 64. To ensure easier convergence, we initialize our model with the 3D geometric and visual prior from Trellis (Xiang et al., 2025).

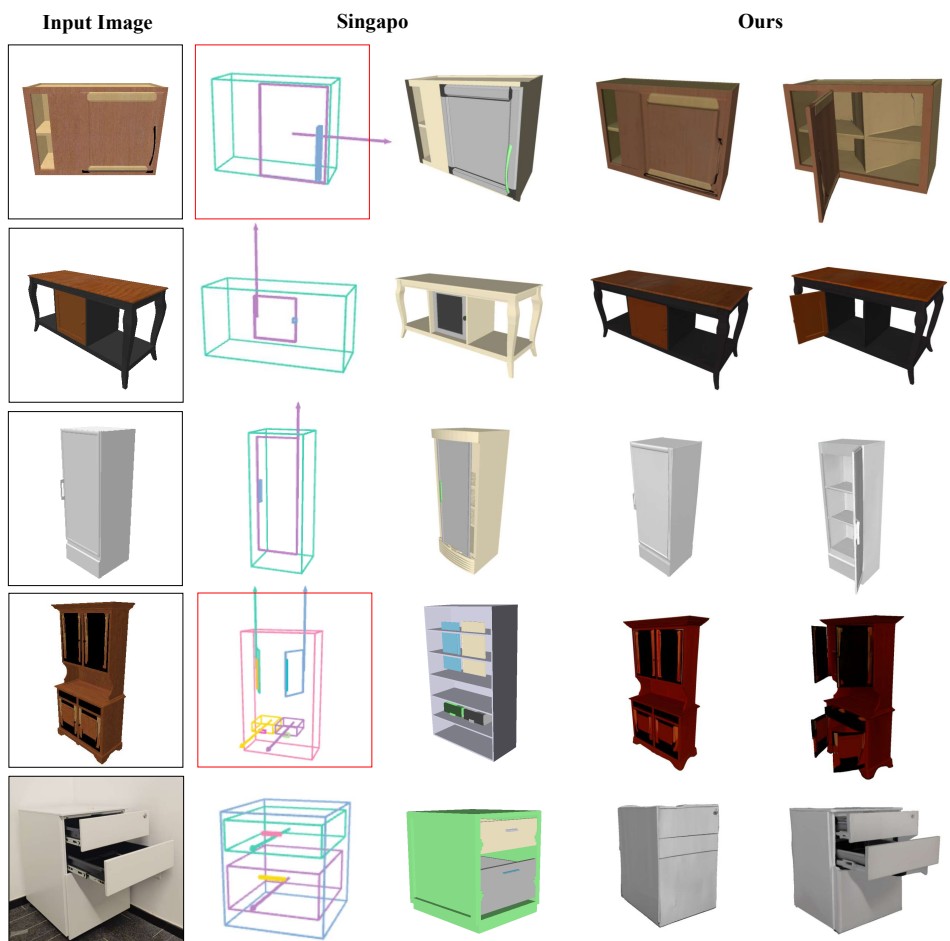

Figure 3: Qualitative results of UniArt. Since retrieval-based methods lack appearance information, we randomly applied different colors to distinguish each link. Our method exhibits better consistency in both appearance and geometry, while the results of Singapo (Liu et al., 2024a) suffer from articulation error (Red Box), geometry inconsistency, and appearance inconsistency.

Table 1: Comparison of generation quality and graph prediction accuracy on **PartNet-Mobility** test set. "-" represents that the code is not available at present.

| Method | Appearance | | Gemoetry | | Shape-image Alignment | |
|---|---|---|---|---|---|---|
| | RS-$d_{PSNR}$ ↑ | AS-$d_{PSNR}$ ↑ | RS-$d_{CD}$ ↓ | AS-$d_{CD}$ ↓ | RS-$d_{OpenShape}$ ↑ | AS-$d_{OpenShape}$ ↑ |
| URDFormer Chen et al. (2024b) | 12.31 | 10.45 | 0.4417 | 0.6910 | 0.0431 | 0.0374 |
| NAP-ICA Lei et al. (2023) | 14.27 | 12.74 | 0.0209 | 0.3473 | 0.0932 | 0.0872 |
| SINGAPO Liu et al. (2024a) | 17.16 | 13.90 | 0.0191 | 0.1270 | 0.1073 | 0.0915 |
| DIPO Shen et al. (2025) | - | - | 0.0132 | 0.0423 | - | - |
| **UniArt (Ours)** | **28.52** | **23.77** | **0.0095** | **0.0376** | **0.1457** | **0.1176** |

## 5.2 EVALUATION METRICS

Previous works on articulated object generation primarily evaluate articulation accuracy. Most benchmarks assume that the geometry of retrieved parts is correct, and therefore ignore two essential aspects: visual fidelity and shape-image consistency, which affect perceptual quality in graphics and simulation.

As a result, existing evaluation protocols underestimate the challenges faced by generative models that must directly synthesize geometry, appearance, and kinematics. To provide a fair and comprehensive benchmark for generation-based methods, we introduce novel evaluation metrics based on the 3D generation task. We follow the common practice of 3D generative models and utilize PSNR,

Table 2: Ablative results of generation quality and articulation prediction on Partnet-Mobility dataset.

| Settings | | Appearance | | Gemoetry | | Shape-image Alignment | |
|---|---|---|---|---|---|---|---|
| Uni-encoding | 3D Prior | RS-$d_{PSNR}$ ↑ | AS-$d_{PSNR}$ ↑ | RS-$d_{CD}$ ↓ | AS-$d_{CD}$ ↓ | RS-$d_{OpenShape}$ ↑ | AS-$d_{OpenShape}$ ↑ |
| ✓ | | 13.24 | 11.76 | 0.0572 | 0.0763 | 0.0504 | 0.0427 |
| | ✓ | 23.75 | 21.37 | 0.0149 | 0.0162 | 0.1175 | 0.1043 |
| ✓ | ✓ | **28.52** | **23.77** | **0.0095** | **0.0376** | **0.1457** | **0.1176** |

Chamfer Distance, and OpenShape (Liu et al., 2023b) metrics to respectively measure the appearance, geometry, and shape-image alignment between generated meshes and conditional input images. All metrics are computed over both resting states and articulated states, denoted as (RS-) and (AS-). For articulated states, we uniformly sample from the resting to the end state and compute the average metrics, following Liu et al. (2024a).

## 5.3 MAIN RESULTS

We report quantitative comparisons on the PartNet-Mobility test set in Table 1. The results demonstrate that our model, UniArt, consistently outperforms prior methods across all metrics, validating the effectiveness of our unified voxel–articulation representation and diffusion-based generation. It is important to note that retrieval-based methods produce uncolored meshes. Thus, we assign ground-truth materials to the uncolored meshes, ensuring a consistent comparison across all models.

In the resting state (RS-), UniArt achieves a PSNR of 28.52, improving over the SINGAPO (Liu et al., 2024a) by 11.36, reflecting highly faithful texture reconstruction. With a Chamfer Distance of 0.0095, our method surpasses DIPO (Shen et al., 2025), highlighting superior fidelity in static shape generation. UniArt also obtains the highest OpenShape score, showing better alignment between generated shapes and conditional input images.

Across articulated states (AS-), performance gains remain substantial, where UniArt reaches 23.77, 9.87 higher than SINGAPO. This demonstrates robustness in preserving appearance even under large part motions. The Chamfer Distance and OpenShape similarity also outperform previous works, setting new state-of-the-art on articulated object generation.

UniArt consistently achieves the best results in terms of appearance fidelity, geometric accuracy, and perceptual alignment. The gains in articulated states are particularly notable, showing that our unified voxel–articulation latent representation ensures stable geometry and motion consistency throughout the articulation process. The qualitative results are shown in Fig. 3.

## 5.4 OPEN-SET EVALUATION

To further verify the generalization capability of UniArt, we evaluate our model on unseen object categories from the PartNet-Mobility benchmark. Specifically, we exclude categories such as Toilet, Laptop or TrashCan during training and only use them for testing. This setting poses a more challenging scenario since the model must synthesize both appearance and kinematics for categories not observed in the training set.

We show the results in Fig. 3. We can see that despite some minor errors, UniArt successfully generates realistic and coherent articulations for the unseen categories, maintaining plausible motion patterns and detailed appearances. Despite never encountering the articulation pattern of these objects during training, the model demonstrates strong generalization by accurately synthesizing their structural parts and corresponding kinematics. This confirms UniArt's capability to handle diverse object categories in an open-set scenario, highlighting its robustness and flexibility for practical applications.

## 5.5 ABLATION STUDY

We conduct ablation studies to analyze the contribution of different components in UniArt. We only report the joint evaluation metrics for the page limit.

**Effectiveness of Uni-encoding of Geometry and Articulation.** UniArt aggregates information from geometry and articulation branches through a sparse structure attention to enforce joint geometry-

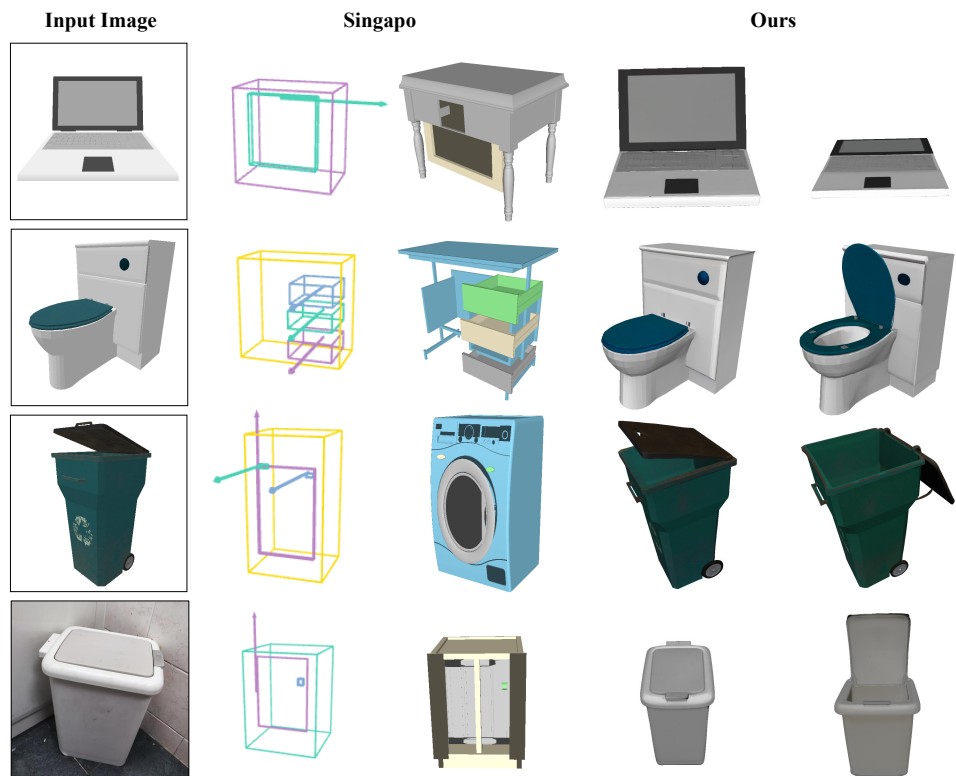

| Input Image | Singapo | Ours |
|---|---|---|

Figure 4: Qualitative results on unseen categories. It can be observed that the articulated objects generated by our method exhibit good consistency with the input images in both appearance and geometry, while previous retrieval-based methods fail to generate sound results.

articulation consistency. In this section, we explore an alternative aggregation strategy, vallina aggregation, where we concatenate the features and then utilize convolution layers to ensure dimension consistency.

As shown in Tab. 2, the vanilla concatenation approach yields a drop in generation quality (RS-$d_{\text{PSNR}}$ decreases from 28.52 to 23.75 and AS-$d_{\text{PSNR}}$ decreases from 23.77 to 21.37), indicating that simple channel stacking fails to align part-specific geometry and articulation information. By contrast, our sparse structure attention design makes a good consistency between articulation and geometry.

**Effectiveness of 3D Shape Prior.** In our implementation, we utilize a shape prior trained from large-scale 3D generative models to make easier modeling of 3D shapes and help prevent unrealistic geometries. In this section, we remove the pretrained 3D shape prior and train UniArt purely from scratch on PartNet-Mobility. We can see from Tab. 2 that all of the metrics degrade significantly. This demonstrates that leveraging a large-scale 3D prior is crucial for stabilizing geometry–articulation interactions.

## 6 CONCLUSION

In this paper, we addressed the challenge of generating articulated objects with coherent geometry, part decomposition, and functional articulation. Existing methods often rely on retrieval-based pipelines, which lead to geometry mismatches and limited category coverage. To overcome these limitations, we proposed UniArt, an end-to-end diffusion-based framework that unifies geometry generation, part segmentation, and URDF prediction into a single model. By formulating segmentation and articulation inference as open-set tasks, UniArt is capable of generalizing to unseen categories and capturing diverse part structures. Extensive experiments on PartNet-Mobility benchmarks demonstrated that our approach significantly outperforms existing baselines, both in mesh fidelity and articulation accuracy, particularly under open-set evaluation.

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

# 7 APPENDIX

## 7.1 APPLICATION

To validate the practical value of UniArt beyond offline metrics, we deploy the generated articulated assets in both a physics simulator and a real-world robotic manipulation setup. Specifically, in the simulator, we export each mesh together with its predicted URDF directly into MuJoCo and PyBullet, where they are instantiated without any manual post-processing; a scripted impedance controller then executes three canonical primitives, hinge opening, slider pulling, and compound flip-and-rotate motions,while success is recorded when the commanded joint approaches at least 70% of its predicted range without self-collision. In the real-robot experiments, we use a 3D Printer to print the generated part and assemble them according to the URDF file. Then, we use an open-source articulation manipulation policy to open the generated objects. The results shown in the Fig. 5 proves the effectiveness of our method.

Generated Articulated Objects          Robot Manipulation of Generated Articulated Objects

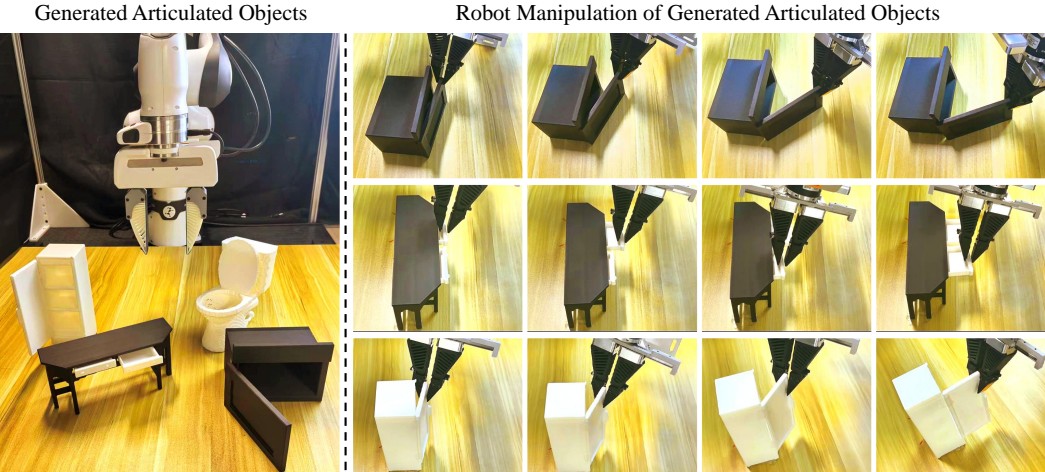

Figure 5: Application in the robotic manipulation.

# 8 LLM USAGE

Large Language Models (LLMs) were used to aid in the writing and polishing of the manuscript. Specifically, we used an LLM to assist in refining the language, improving readability, and ensuring clarity in various sections of the paper. The model helped with tasks such as sentence rephrasing, grammar checking, and enhancing the overall flow of the text.

It is important to note that the LLM was not involved in the ideation, research methodology, or experimental design. All research concepts, ideas, and analyses were developed and conducted by the authors. The contributions of the LLM were solely focused on improving the linguistic quality of the paper, with no involvement in the scientific content or data analysis.

The authors take full responsibility for the content of the manuscript, including any text generated or polished by the LLM. We have ensured that the LLM-generated text adheres to ethical guidelines and does not contribute to plagiarism or scientific misconduct.

