# OpenReview forum: "UniArt: Generating 3D articulated objects with open-set articulation beyond retrieval"
_ICLR.cc/2026/Conference — ICLR 2026 Conference Withdrawn Submission_

### Official Review · Reviewer_9dVL · 2025-10-25

**Soundness:** 2
**Presentation:** 2
**Contribution:** 3
**Rating:** 4
**Confidence:** 3

**Summary:**

This paper proposes UniArt, a framework that directly uses diffusion models to generate 3D articulated objects using a single RGB image. The proposed framework achieves this by converting an articulated object (mesh + URDF) into a single latent vector named UniLatent, which "jointly encodes geometry, appearance, part structure, and the articulation tree", and using diffusion models for generation.

**Strengths:**

1. The focus of this paper, namely the generation articulated objects directly using generative models, is a meaningful problem of great practical value.
2. The figure illustrations are visually well-designed and pleasing to watch.
3. The experiments give promising results.

**Weaknesses:**

1. The presentation of the method is relatively vague without many details. While it can be figured out that the paper is trying to map articulated objects into latent vectors with VAEs and use diffusion models for generation in the latent space. There is not enough information to learn about the specific steps. For example, how is the articulation (joints) information projected to voxel representations? And how is a UniLatent vector decoded to a URDF with meshes? More are listed in the "Questions" section. It is also suggested that the authors add detailed illustrations for the proposed approach.
2. While the paper claims to generate robot-ready articulated objects, there is no robot-related experiments (e.g. object manipulation) with quantitative or qualitative results to support this claim.
3. (minor) Typos, for example, "$u_i$" in L229 (which should be $u^i$?), "mod-el" in L305.

**Questions:**

1. In L236-237, the authors use a 9-dimensional vector $c$ to represent the URDF parameters, regardless of its size. How is this achieved? And why use a 9-dimensional vector instead of other dimensionalities?
2. In Eq.10, how is each term of the loss function calculated?

---

### Official Review · Reviewer_qEA2 · 2025-10-25

**Soundness:** 2
**Presentation:** 2
**Contribution:** 3
**Rating:** 4
**Confidence:** 3

**Summary:**

UniArt is an end-to-end diffusion framework that generates 3D articulated objects directly from a single image, without relying on predefined part libraries. It unifies geometry, segmentation, and joint parameters in a shared latent space and models joint prediction as an open-set problem, enabling generalization to unseen categories. Experiments show that UniArt surpasses retrieval-based methods in generation quality, geometric consistency, and joint accuracy, with results readily usable for robotic simulation and manipulation.

**Strengths:**

-   It proposes a non-retrieval paradigm for articulated object generation without relying on predefined part libraries, representing a novel and innovative approach.
-   UniArt formulates part segmentation and joint type prediction as open-set problems, meaning they do not rely on predefined part or joint semantic labels from the training set. This design enables the model to generalize to unseen object categories.
-   Experiments demonstrate the effectiveness of UniArt in geometric fidelity and generation quality.
-   The generated models can be directly deployed in simulation environments, facilitating downstream robotic applications and practicality.

**Weaknesses:**

-   Table 2 shows a significant performance drop without the 3D Prior, indicating that UniArt’s stability and generalization may largely rely on external 3D knowledge, with limited clarification provided.
-   Experiments
    -   The experiments mainly compare UniArt with retrieval-based methods, lacking comparisons with non-retrieval approaches to better demonstrate the advantages of the proposed framework.
    -   The paper only states that the main experiments involve "unseen categories (Bottles, Toilet, Chair, etc.)", but does not specify unseen categories, making it difficult to assess fairness and reproducibility.
    -   Figure 4 is not referenced in the paper. It may have been intended to replace "Fig.3" in Line 419.
    -   Figure 4 uses Singapo as a baseline for qualitative comparison. However, Singapo fails to handle unseen object categories and makes incorrect structural predictions, making the comparison less meaningful.
-   Appendix 7.1. Although the paper claims that the generated articulated objects are applied in simulation and real-world scenarios through 3D printing, several points remain unclear:
    -   The paper does not describe what the "open-source articulation manipulation policy" refers to or how it is implemented.
    -   The paper lacks experimental videos in simulation or real-world settings to intuitively demonstrate the accuracy and physical plausibility of generated joints in simulation or by 3D printing.
-   Failure Analysis. The paper presents successful examples but lacks a systematic analysis of failure cases. It remains unclear on which object types or scenes the model tends to fail, and whether the failures stem from complex joint structures, severe occlusions, or limited open-set generalization.
-   Minor typos. *e.g.* "Experimical Setup" in title of section 5.1, "Gemoetry" in Table 1.

**Questions:**

-   The paper embeds joints into voxel space via "joint-to-voxel projection". How does this approach accurately encode a sparse joint axis within a dense voxel grid, and robustly recover it from potentially noisy voxel features during decoding?
-   Could the authors provide additional qualitative comparisons with other baselines, such as URDFormer or MeshArt?
-   In URDF models, joint rotations and translations require no specific physical mechanical structures, whereas real-world objects do. This creates a gap between directly 3D-printing URDF models and building functional articulated objects in real world. How do the authors fix this problem to 3D-print the models in real world?
-   Could the authors provide more details on the experiments, such as the data settings?

---

### Official Review · Reviewer_knsF · 2025-11-01

**Soundness:** 1
**Presentation:** 1
**Contribution:** 2
**Rating:** 2
**Confidence:** 4

**Summary:**

This paper approaches the task of articulated object generation from the perspective of conditional generation. It develops a diffusion-based framework and proposes a complete algorithmic pipeline. In experiments, the proposed method is compared with several retrieval-based approaches and achieves competitive results. Notably, the paper formulates articulated object generation as an open-set problem, enabling cross-category generation.

**Strengths:**

1. This work raises a highly meaningful problem, moving the task of articulated object generation beyond retrieval-based approaches, thereby enabling the generation of more plausible and diverse results.
2. This paper formulates articulated object generation as an open-set problem, allowing the model to generate object categories unseen during training. Moreover, the visualized results are impressive.

**Weaknesses:**

1. The first claimed contribution — “We reformulate the articulated object creation task as a conditional generation task” — does not appear to be particularly novel. Similar ideas of conditional generation have already been explored in prior works such as MeshArt[1] and ArtFormer[2].
2. The experimental setup of this paper does not effectively validate the advantages of the proposed method. In the introduction, the authors highlight several limitations of retrieval-based approaches — namely geometric misalignment, limited diversity, and poor generalization to objects outside the training distribution. However, the three metrics used in their experiments mainly measure the similarity between the generated objects and the image conditions, which cannot adequately demonstrate that their method addresses the issues of geometric misalignment and limited diversity.
3. The paper only compares its method with retrieval-based approaches, while several recent works on articulated object generation have already moved beyond retrieval-based frameworks. Without comparisons to these more recent non-retrieval methods, the paper fails to convincingly demonstrate the superiority of its approach within the broader articulated object generation domain.
4. In the experiments, the generated results from this paper contain textures, while the compared methods do not have the capability to generate textures. Therefore, comparing visual quality metrics between the two is not fair. Moreover, the method section does not clearly explain how textures are generated in this work, which raises further questions about the evaluation and implementation details.
5. There are numerous errors throughout the paper, including frequent typos and misreferenced figures.

[1] Daoyi Gao, Yawar Siddiqui, Lei Li, and Angela Dai. Meshart: Generating articulated meshes with structure-guided transformers. In Proceedings of the Computer Vision and Pattern Recognition Conference, 2025. 1

[2] Jiayi Su, Youhe Feng, Zheng Li, Jinhua Song, Yangfan He, Botao Ren, and Botian Xu. Artformer: Controllable generation of diverse 3d articulated objects, 2025. 1

**Questions:**

1. What is the main problem this paper aims to solve — generating a large number of highly diverse articulated objects, or controllable generation based on a single image? The paper does not clearly articulate its motivation, leaving readers uncertain about the core objective of the work.
2. Why didn’t the paper use commonly adopted evaluation metrics in articulated object generation, such as MOD, COV, and FID, which could provide a more objective assessment of visual and geometric quality?
3. The paper compares its method with several others in the experiments, but why does it only present qualitative results for a single method?
4. How are the textures in the generated articulated objects obtained? Why are previous methods unable to generate textures? What are the key factors that enable your method to produce textured results?
5. The proposed method removes the semantic labels of each part to better generalize to unseen categories. Shouldn’t this claim be supported by experiments? For example, would the generalization ability disappear if semantic labels were added?

---

### Official Review · Reviewer_wxyC · 2025-11-01

**Soundness:** 2
**Presentation:** 2
**Contribution:** 2
**Rating:** 2
**Confidence:** 3

**Summary:**

This paper presents UniArt, an end-to-end diffusion-based framework for generating 3D articulated objects directly from a single image. Unlike prior retrieval-based pipelines that assemble parts from existing datasets, UniArt jointly synthesizes geometry, part segmentation, and articulation parameters within a unified voxelized latent space. The proposed approach introduces a joint-to-voxel embedding that spatially aligns kinematic parameters with geometric occupancy. Furthermore, the authors treat articulation prediction as an open-set problem, allowing the model to generalize beyond predefined joint semantics.

**Strengths:**

- Novel unified generation framework:
  The paper introduces an end-to-end diffusion-based architecture that jointly models shape and motion, which is a clear step forward from retrieval-based pipelines.
- Open-set articulation modeling:
  Reformulating articulation type prediction as an open-set problem is conceptually sound and improves generalization to unseen joint types or object categories.
- Potential impact:
  The method contributes to scalable creation of articulated assets for robotics, simulation, and embodied AI, aligning with current trends in large-scale 3D generation.

**Weaknesses:**

- Overclaimed contribution.
  The first claimed contribution — “We reformulate the articulated object creation task as a conditional generation task” — appears overstated. Previous works have already explored conditional generation of articulated objects from text or images (e.g., [ArtFormer](https://arxiv.org/abs/2412.07237)). Therefore, this claim does not seem entirely novel.
- Retrieval-based comparison inconsistency.
  In the introduction, the authors emphasize that retrieval-based methods suffer from limited diversity. However, this claim is not supported by quantitative evaluation in later experiments. There is no metric or experiment explicitly measuring diversity or variability in the generated shapes.
- Qualitative comparisons are incomplete.
  Although quantitative results include multiple baselines (e.g., DIPO, NAP-ICA), the qualitative figures only compare UniArt with SINGAPO. This limits the reader’s ability to visually assess improvements over other methods.
- Questionable metric setup.
  The evaluation metrics may be biased toward UniArt. Specifically: 1. Other methods generate untextured meshes, while PSNR measures image-level fidelity—potentially unfair to texture-free baselines. 2. For geometry evaluation, the paper does not include widely adopted metrics such as MMD (Minimum Matching Distance) and COV (Coverage), which are standard for generative 3D model assessment. 3. Despite repeatedly mentioning improved articulation accuracy in the abstract and conclusion, the paper does not report articulation accuracy as a quantitative metric.

**Questions:**

- Metrics:
  Are there more suitable or meaningful metrics for evaluating articulated object generation, especially regarding articulation correctness and geometric fidelity?
- Method clarity:
  In Section 4.3, the generation equations seem to omit the image conditioning term. Could the authors clarify how the input image is incorporated into the generation process?
- Retrieval-based comparison:
  The paper critiques retrieval-based methods for limited diversity, yet this aspect is not quantitatively compared. Could the authors clarify how diversity was measured, or provide an evaluation that supports this claim?
- Texture generation:
  The qualitative results shown in the paper include textured meshes; however, the methodology section does not explain how texture generation is handled. Could the authors clarify whether the texture is generated jointly with geometry, transferred from the input image, or applied through a separate process?

---

### Note · Authors · 2025-11-15

I have read and agree with the venue's withdrawal policy on behalf of myself and my co-authors.